# Transmission of 'Candidatus Anaplasma camelii' to mice and rabbits by camel-specific keds, *Hippobosca camelina*

Joel L. Bargul[1,2]*, Kevin O. Kidambasi[1,2], Merid N. Getahun[1], Jandouwe Villinger[1], Robert S. Copeland[1], Jackson M. Muema[1,2], Mark Carrington[3], Daniel K. Masiga[1]

**1** International Centre of Insect Physiology and Ecology (*icipe*), Nairobi, Kenya, **2** Department of Biochemistry, Jomo Kenyatta University of Agriculture and Technology (JKUAT), Nairobi, Kenya, **3** Department of Biochemistry, University of Cambridge, Tennis Court Road, Cambridge, United Kingdom

\* jbargul@icipe.org

**Data Availability Statement:** All nucleotide sequence files are available from the NCBI GenBank database with accession numbers as

## Abstract

Anaplasmosis, caused by infection with bacteria of the genus *Anaplasma*, is an important veterinary and zoonotic disease. Transmission by ticks has been characterized but little is known about non-tick vectors of livestock anaplasmosis. This study investigated the presence of *Anaplasma* spp. in camels in northern Kenya and whether the hematophagous camel ked, *Hippobosca camelina*, acts as a vector. Camels ($n = 976$) and > 10,000 keds were sampled over a three-year study period and the presence of *Anaplasma* species was determined by PCR-based assays targeting the *Anaplasmataceae* 16S rRNA gene. Camels were infected by a single species of *Anaplasma*, 'Candidatus Anaplasma camelii', with infection rates ranging from 63–78% during the dry (September 2017), wet (June-July 2018), and late wet seasons (July-August 2019). 10–29% of camel keds harbored 'Ca. Anaplasma camelii' acquired from infected camels during blood feeding. We determined that *Anaplasma*-positive camel keds could transmit 'Ca. Anaplasma camelii' to mice and rabbits via blood-feeding. We show competence in pathogen transmission and subsequent infection in mice and rabbits by microscopic observation in blood smears and by PCR. Transmission of 'Ca. Anaplasma camelii' to mice (8–47%) and rabbits (25%) occurred readily after ked bites. Hence, we demonstrate, for the first time, the potential of *H. camelina* as a vector of anaplasmosis. This key finding provides the rationale for establishing ked control programmes for improvement of livestock and human health.

## Author summary

Hematophagous flies such as Tabanids and *Stomoxys*, and other biting flies, are mechanical transmitters of various pathogens such as African trypanosomes and *Anaplasma* species. However, little is known about the role of common camel-specific biting keds (also known as camel flies or louse flies, genus *Hippobosca*) in pathogen transmission. Keds inflict painful bites to access host blood, and in the process may transmit bacterial hemopathogens. We demonstrated using amplicon sequencing that camel keds can transmit

follows; (i) Camel keds: MK754149-MK754151 and MT510535-MT510537, (ii) Camels: MK754152-MK754154, MT510527, MT510529, MT510531, MT510532, and MT510534, (iii) Mice: MK754155-MK754160 and MT510538, and (iv) Rabbit: MT510539. The GenBank accessions for the longer ~1000-bp sequences used for phylogenetic analysis include; MK388294-MK388300, MT510528, MT510530 and MT510533 (sequenced in camel) & MK388301 (sequenced in test mouse after ked feeding bites).

**Funding:** This work was supported through the DELTAS Africa Initiative grant # DEL-15-011 to THRiVE-2 (awarded to JLB). The DELTAS Africa Initiative is an independent funding scheme of the African Academy of Sciences (AAS)'s Alliance for Accelerating Excellence in Science in Africa (AESA) and supported by the New Partnership for Africa's Development Planning and Coordinating Agency (NEPAD Agency) with funding from the Wellcome Trust grant # 107742/Z/15/Z and the UK government. This research study was also supported by funding from the International Foundation for Science (IFS), Stockholm, Sweden, through an IFS grant # B/5925-1 to JLB. Further funding for this work was received from Cambridge-Africa Alborada Fund (#RG86330) and The Global Challenges Research Fund (GCRF grant #G100049/17588) both awarded to MC and JLB. MC is a Wellcome Trust Investigator (217138/Z/19/Z). The International Centre of Insect Physiology and Ecology receives support from the: UK's Foreign, Commonwealth & Development Office (FCDO); the Swedish International Development Cooperation Agency (Sida); the Swiss Agency for Development and Cooperation (SDC); the Federal Democratic Republic of Ethiopia; and the Government of the Republic of Kenya. The views expressed herein do not necessarily reflect the official opinion of the donors. The funders had no role in study design, data collection and analysis, decision to publish, or preparation of the manuscript."

**Competing interests:** The authors have declared that no competing interests exist.

'*Candidatus* Anaplasma camelii' from naturally-infected camels to healthy mice and rabbits via blood feeding. The high prevalence of camel anaplasmosis throughout the year in northern Kenya could be attributed to keds, which infest camels all year round. Unlike ticks, keds can fly from one host to another and thus promote disease transmission among and between camel herds. Although our study focused on the transmission of *Anaplasma* sp. by camel keds, these flies could possibly also transmit other hemopathogens through a similar mechanism. Notably, in the absence of their preferred hosts, keds occasionally bite humans and other vertebrates they come across in search of bloodmeals, and in the process could transmit zoonotic pathogens.

## Introduction

*Anaplasma* species are obligate rickettsial pathogens that proliferate inside red blood cells and cause anaplasmosis in domestic and wild animals. Various *Anaplasma* species such as *Anaplasma marginale*, *A. centrale*, *A. platys*, *A. bovis*, *A. ovis*, and *A. phagocytophilum* cause huge agricultural losses by severely constraining livestock production and in addition can cause zoonotic infections of humans [1,2]. Clinical signs of anaplasmosis during acute infection include anaemia, pyrexia, reduced milk yield, loss of body condition, abortion, and death [3–5].

*Anaplasma* is transmitted by different species of hard ticks [6] as well as mechanically via contaminated mouthparts of *Stomoxys calcitrans* and Tabanidae, among other biting flies, albeit with reduced efficiency [7,8]. Additionally, pathogen transmission can occur via needles and other veterinary instruments contaminated with fresh infected blood [9]. Mechanical transmission of *Anaplasma* pathogens is thought to be possible only at high parasitaemia [7].

There are increasing reports of camel anaplasmosis with pathogens that include '*Candidatus* Anaplasma camelii' and the related dog pathogen, *Anaplasma platys* [10–12]. Further, *A. platys* continues to be detected in other vertebrate hosts, including humans [13–15], sheep [16], cattle [17], and cats [18]. Close association between humans and their livestock, and co-herding of domestic animals, often in close proximity with wildlife, amplifies chances of spreading vector-borne diseases [15,19]. In Kenya, there are currently 3.34 million camels [20] but the veterinary and zoonotic importance of anaplasmosis in this large livestock resource is not well understood.

Camels are kept by nomadic pastoralists in northern Kenya for milk, meat, hides, transport, income from selling milk, and for social capital. Since camels survive readily in harsh arid and semi-arid regions, they are the preferred livestock. However, their productivity is constrained by several factors but chiefly pests and diseases. Other than ticks, keds are the major hematophagous ectoparasitic camel pests and can be found to infest 100% of camel herds throughout the year in northern Kenya [21]. Like other hippoboscids, both adult male and female keds are obligate blood feeders that associate closely with their camel hosts. They can freely move from one camel host to the next, within or between herds, often when disturbed as the host responds to painful blood-feeding bites. This could facilitate transfer of pathogens from one animal to another. Keds mainly infest their vertebrate hosts' underbelly, although they can be found on the other parts of the body such as neck, ears, hump, and girth [21]. In addition to annoyance and the painful bites they inflict while feeding, keds contribute to anaemia and reduced milk and meat production in camels [22,23]. Keds are members of Hippoboscidae, which includes tsetse flies; members of this family act as vectors of infectious agents such as protozoa [24], bacteria [25], filarial nematodes [26], and viruses [27]. However, camel keds lack competence to transmit *Trypanosoma evansi* to mice [23], and cattle keds, *Hippobosca*

*rufipes*, have been found incompetent to transmit *Anaplasma marginale* from infected cattle to experimental oxen [28]. The main goal of this study was to determine the vector competence of camel keds in transmission of *Anaplasma* species.

## Materials and methods

### Study area

This study was carried out in Laisamis Ward (1.6˚N 37.81˚E, 579 m ASL) located in the south of Marsabit County. Sampling sites were selected along two seasonal rivers that provide drinking water for livestock, namely: Laisamis River in Laisamis and Koya River located about 29 km south east of Laisamis town (N 01˚ 23' 11"; E 37˚ 57' 11.7", 555 m ASL) (Fig 1). Marsabit County had a population of approximately 203,320 camels in 2017 with 86% of the household heads, whose major occupation was livestock herding, deriving their livelihoods from the sale of livestock [29].

### Weather conditions

The study region is arid and semi-arid [21]. Due to climate variability, protracted dry seasons are common resulting in depletion of pastures, decreased livestock productivity, and livestock death [29]. Vegetation dries up soon after wet season except some drought-resistant evergreen trees and shrubs, such as *Acacia tortilis*, *Cordia sinensis*, *Salvadora persica*, *Euphorbia tirucalli*, among others, that camels feed on in the dry season.

### Study design and sample collection

This study was cross-sectional in design. Camel blood and ked (*Hippobosca camelina*) samples were randomly collected on daily basis through opportunistic sampling from the available

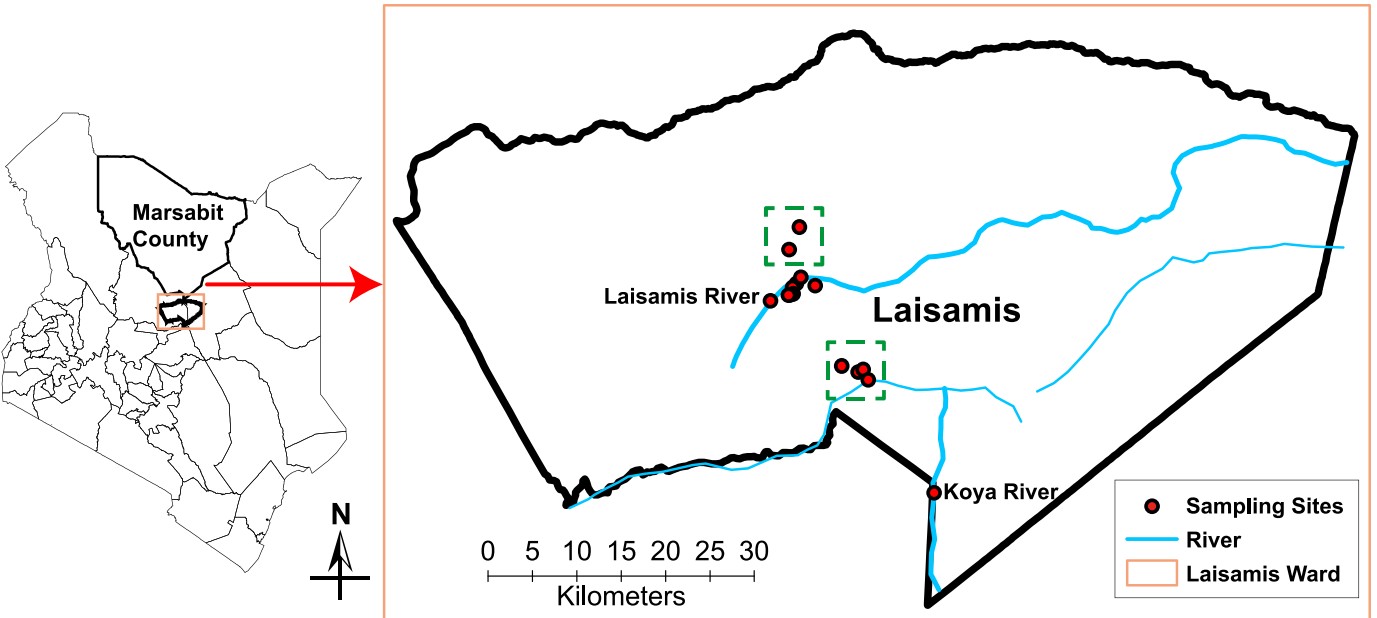

**Fig 1. The map of Kenya showing the sampling sites in Laisamis in Marsabit County.** The following shapefiles were utilised on the map together with the websites where the data was sourced: Kenya administrative boundaries– https://africaopendata.org/dataset/kenya-counties-shapefile; License: https://www.opendefinition.org/licenses/cc-by. Rivers–(**ke_major-rivers.zip**) https://www.wri.org/resources/data-sets/kenya-gis-data; License:https://www.wri.org/publications/permissions-licensing. All the websites used were under the Creative commons BY 4.0. No base map was used to create the map.

camel herds at the geo-referenced sites. This way, we circumvented lifestyle challenges associated with nomadic pastoralism such as frequent long distance migration with camels into remote inaccessible areas in search of pastures.

Camel blood (*n* = 249) and keds were first sampled during the dry season in September 2017. Additional camels (*n* = 280) were sampled for blood and keds during the late wet season in June-July 2018 and in July-August 2019 (*n* = 447 camels).

We lacked information on *Anaplasma* prevalence in Laisamis sub-County for reference in calculating sample sizes due to paucity of historical data, such as on the current camel population and production or the burden of pests and diseases. Therefore, we aimed at collecting as many samples as possible during the three main field visits that each lasted between 7–30 days, and we sampled all camels and keds in randomly selected herds. A herd was defined as a group of camels living together as a unit, often feeding and migrating together.

## Ethics statement

We collected camel blood and camel ked samples, and conducted pathogen transmission experiments in mice and rabbits with strict adherence to the experimental guidelines and procedures approved by the Institutional Animal Care and Use Committee (IACUC) at the International Centre of Insect Physiology and Ecology (*icipe* IACUC REF: IACUC/ICIPE/003/ 2018). Camels, mice and rabbits were handled carefully to ensure minimum distress. Livestock keepers were informed about the study prior to sample collection following verbal consent as many camel herders were unable to read or write.

## Camel blood collection

The following information was recorded during collection of blood samples; sex, age, count of infesting keds, pregnancy and abortion history, and assessment of the body condition. Camels were sampled for collection of 5 mL of blood via jugular venipuncture. Blood was drawn into 10 mL EDTA vacutainer tubes (Plymouth PLG, UK) and kept under cold chain at 4˚C during handling process. Each labeled sample was transferred into a 2-mL cryotube (Greiner Bio-one, North America, Inc.) and then preserved in liquid nitrogen for transportation to *icipe* laboratories for molecular analysis to detect pathogens.

## Collection of camel keds, *Hippobosca camelina*

We observed that at daytime, collection of keds from camels by hand was difficult because the flies responded quickly by moving away, often landing on the same or nearby camel. In contrast, keds seemed to be docile at night, thus capturing them was easy. Therefore, whereas blood samples were obtained during the day when camels converged at the watering points along the rivers, keds infesting them were collected later at night between 20:00–02:00 hrs. Use of sweep nets for fly collection was discontinued because it frightened camels. Thus, attached keds were hand-picked from camels by 3–4 trained field assistants. To locate keds on the camel, a spotlight was briefly switched on and then off to minimize light-induced activity (movement) in keds.

Camel keds were strictly collected only from the blood-sampled camel herds to allow for comparison of the hemopathogens occurring in camels and their keds. The keds were preserved either in the liquid nitrogen or kept at room temperature in absolute ethanol.

## Survey of camel keds; fly populations and sex

To determine the seasonal changes in the average numbers of camel-infesting flies, live keds on camels were counted during the daytime using 8-key manual differential counter (Fisher

Scientific, USA) during dry and wet seasons. Camel keds were collected from camel herds and preserved in absolute ethanol and transported to the laboratory for identification. The flies were sorted by sex. Keds were also randomly collected from donkeys, cattle, goats, dogs, and sheep for morphological and molecular identification to establish whether *H. camelina* infests other livestock species.

## Morphological identification of camel keds

Keds were morphologically identified at the Zoology Museum of the University of Cambridge (UK) and the Natural History Museum in London using standard morphological keys and by comparison to the preserved ked collections [30].

## Transmission of '*Ca*. Anaplasma camelii' from camels to mice and rabbits by camel keds

The ability of camel keds, *H. camelina*, to transmit *Anaplasma* spp. to mice and rabbits was studied by following published protocols with minor modifications [23]. Camel keds were collected from camels and placed into well-ventilated cages covered with black netting. The keds were then released into larger cubical cages measuring $30 \times 30 \times 30$ cm containing restrained mice for bloodmeal acquisition. Rabbits were shaved to reduce hair on their back to facilitate fly feeding. Freshly collected camel keds that had been kept in well-ventilated cages were then placed on the shaved back of rabbits to allow bloodmeal acquisition.

Laboratory-reared Swiss white mice ($n = 21$), maintained under clean conditions for 6–8 weeks, were used in the first pathogen transmission study conducted in April 2018. In July 2018, the transmission study was repeated using immunosuppressed mice ($n = 60$; test = 58, and control = 2) to assess the effect of immunosuppression on pathogen acquisition. A third independent field-based study on pathogen transmission by keds was set up in July-August 2019 using both mice ($n = 123$) and rabbits ($n = 6$). Thus, mice and rabbits were transported to the field sampling sites in northern Kenya to provide blood-meals for keds freshly collected from camels and identify transmitted ked-borne pathogens. Ked feeding schedules were designed as shown in Table A-C in S1 Text. During the third field study in July-August 2019, camel keds were collected from a total of 35 camel herds from seven different rural settlements in Laisamis Ward.

## Exposure of camel keds to mice and rabbits for bloodmeal acquisition with concommittant '*Ca*. Anaplasma camelii' transmission

**Mouse-ked exposure.** Keds collected from camels within the previous 15 min were directly placed into cages measuring $30 \times 30 \times 30$ cm containing restrained mice to continue blood-feeding on mice for 12 h ($n = 20$ keds/mouse). Ked-exposed mice were allowed to rest for 12 h before the next fly feeding. All experiments were conducted in a fly-proof environment inside insecticide-free rectangular mosquito nets to exclude any other biting flies. The control mice were protected from receiving any inadvertent bites from biting flies by further wrapping their cages with fly-proof nets. The third pathogen transmission study was conducted under similar conditions as described before, with minor changes on the host-ked exposure; this time using both mice and rabbits as bloodmeal sources (Table C in S1 Text). A slight modification was adopted to improve feeding success in flies; we placed fly cages, which are covered with a netting on both sides, on the shaved body parts of restrained mice and rabbits, rather than introducing restrained mice into the fly cages.

**Rabbit-ked exposure.** Restrained test rabbits ($n = 4$) were exposed to ked bites to feed repeatedly on each rabbit with a 2-day interval in between successive feeds to increase chances of pathogen transmission, if at all possible. Keds inflicted bites on the rabbits' ears and on the shaved region on the back ($n = 40$ keds/rabbit). The control group ($n = 2$) was protected from biting flies throughout the study.

## Screening of mice and rabbits post-ked bites

For pathogen screening, blood samples were collected from both control and test mice and rabbits after two weeks of ked exposure. Towards the end of the follow up pathogen detection studies, all mice were anaesthetized prior to collection of about 1 mL of blood from the heart using 1-mL syringes containing 200 µL of Carter's Balanced Salt Solution with 10,000 Units/L heparin sodium salt. Blood samples were also obtained from rabbits by pricking the ear vein with a lancet followed by sample collection in heparinized capillary tubes. Wet blood smears were prepared for staining to detect *Anaplasma* infection and total DNA was extracted from blood samples for time-course detection of *Anaplasmataceae* by PCR-based assays, as described below.

## Field's staining of blood smears for parasitological examination for detection of *Anaplasma* spp

Rapid Field's staining of thin blood films was done to detect *Anaplasma* species in mice ($n = 8$) and rabbits ($n = 8$), as well as in the fixed camel smears ($n = 13$) prepared during field sampling. The tips of mouse-tails and rabbit ears were sterilized with 70% v/v ethanol, and veins were pricked using sterile lancets to collect drops of blood on microscope slides for preparation of thin blood smears. Each lancet was used only once to avoid iatrogenic transmission of *Anaplasma* to mice and rabbits.

During field collection of samples, a drop of blood from each camel was placed on to a slide for preparation of thin blood smears. The thin blood smears were labeled and allowed to air dry before fixation in methanol for 1 minute. Field's staining was performed on the air-dried smears by flooding each slide with 1 mL of 20% Field's stain B consisting of methylene blue and Azure dissolved in a phosphate buffer solution. This was immediately followed by addition of equal volume of 100% Field stain A consisting of Eosin Y in a buffer solution. Field's stains A and B were mixed well using Pasteur pipettes and the staining reaction was allowed for 1 minute, followed by rinsing of slides in distilled water. The slides were air-dried for imaging to morphologically identify intracellular species of *Anaplasma* pathogens using digital camera ZEN lite 2012 mounted on to a compound microscope equipped with 60× and 100× oil-immersion objectives [31]. Ten (10) microscopic fields were counted for each slide and average percentage bacteremia levels determined as in formula below;

$$\% \text{ bacteremia} = \frac{\text{Number of infected RBCs per field}}{\text{Total number of RBCs}} \times 100 \qquad (1)$$

## Molecular identification of *Anaplasma* species

**DNA extraction from blood and ked samples.** Each camel ked, with an average body weight of 100 mg, was surface-sterilized with 70% ethanol and briefly allowed to air-dry on a paper towel. Each whole ked was then placed into a clean 1.5-mL microfuge tube containing sterile 250 mg of zirconia beads with 2.0-mm diameter (Stratech, UK) and ground in liquid nitrogen in a Mini-Beadbeater-16 (BioSpec, Bartlesville, OK, USA) for 2 min. One hundred microliters (100 µL) of each blood sample was pipetted into 1.5-mL Eppendorf tubes and mixed with 20 µL proteinase K, and the total volume was adjusted to 220 µL with 1× PBS at

**Table 1. Primers for PCR amplification.**

| Primer name | 5' to 3' sequence | Target organism | Target gene | Amplicon size (bp) | Primer reference |
|---|---|---|---|---|---|
| *Anaplasma*JV-F<br>*Anaplasma*JV-R | CGGTGGAGCATGTGGTTTAATTC<br>CGRCGTTGCAACCTATTGTAGTC | *Anaplasma* spp. | Partial 16S rRNA | 300 | [32] |
| EHR16SD<br>1492R | GGTACCYACAGAAGAAGTCC<br>GGTTACCTTGTTACGACTT | *Anaplasma* & *Ehrlichia* spp. | Longer 16S rRNA | 1000 | [33,34] |
| MSP4-F<br>MSP4-R | CCCATGAGTCACGAAGTGG<br>GCTGAACAGGAATCTTGCTCC | *A. marginale* | Major surface protein 4 | 753 | [35] |

pH 7.4. Total genomic DNA was extracted from individual keds and blood samples from camels, mice, and rabbits using DNeasy Blood & Tissue Kit (Qiagen, Hilden, Germany) following the manufacturer's instructions.

**PCR—HRM, purification of PCR amplicons, and gene sequencing.** We screened for *Anaplasma* spp. by PCR followed by high-resolution melting (PCR-HRM) analysis in an HRM capable RotorGene Q thermocycler (Qiagen, Hannover, Germany) using *Anaplasma*JV-F and *Anaplasma*JV-R primers (Table 1) as previously described [32].

The PCR-HRM assays were carried out in 10-μL reaction volumes containing 2.0 μL of 5× HOT FIREPol EvaGreen HRM mix (no ROX) (Solis BioDyne, Estonia), 0.5 μL of 10 pmol of each primer, 6.0 μL PCR water, and 1.0 μL of template DNA. The amplification conditions included an initial enzyme activation step at 95˚C for 15 min, followed by 10 cycles at 94˚C for 20 sec, step-down annealing for 25 sec from 63.5˚C to 53.5˚C, decreasing the temperature by 1˚C after each cycle, and an extension step at 72˚C for 30 sec, then 25 cycles at 94˚C for 25 sec, annealing at 50.5˚C for 20 sec, and extension at 72˚C for 30 sec, and a final elongation at 72˚C for 7 min. Immediately after PCR amplifications, HRM curves of the amplicons were obtained by increasing temperature gradually from 75˚C to 95˚C at 0.1˚C/2 sec increments between successive acquisitions of fluorescence. Negative and positive controls were included in all PCR-HRM assays. The HRM profiles were assessed using Rotor-Gene Q Series Software 2.3.1 (Build 49). Changes in fluorescence with time (dF/dT) were plotted against corresponding changes in temperature (˚C). PCRs generated 300-bp 16S rRNA gene region amplicons. Representative samples were further amplified by conventional PCR for sequencing. This amplification was performed using EHR16SD and 1492R primers [33,34] (Table 1) targeting ~1000-bp longer fragment of the *Anaplasma* 16S rRNA gene. Longer sequences enabled species identification and were used in phylogenetic analysis.

The PCRs were performed in 15-μL reaction volumes that included 5.0 μL PCR water, 1.0 μL of DNA template, 3.0 μL of 5x HOT FIREPol Blend Master Mix (Solis Biodyne, Estonia), and 0.5 μL of 10 μM EHR16SD and 1492R primers (Table 1). The cycling conditions consisted of: 95˚C for 15 min; two cycles of 95˚C for 20 sec, 58˚C for 40 sec, and 72˚C for 90 sec; three cycles of 95˚C for 20 sec, 57˚C for 30 sec, 35 cycles of 95˚C for 20 sec, 56˚C for 40 sec and 72˚C for 80 sec, and a final extension at 72˚C for 10 min. The amplifications were carried out in a ProFlex PCR system (Applied Biosystems by life technologies).

PCR amplicons were resolved on 1% ethidium bromide-stained agarose gels run at 80 V for 1 hr. The DNA was visualized under UV transillumination and the target bands were excised and purified from the gels using QIAquick PCR purification kit (Qiagen, Valencia, CA) following manufacturer's instructions, and Sanger sequenced (Macrogen Europe, Amsterdam).

## Data analysis

A map of the sampling locations was created using geo-referenced data uploaded on the ArcMap extension of ArcGIS v 10.5 software by Esri (https://desktop.arcgis.com/en/arcmap/10.5/).

Data on ked catches was entered into Microsoft Excel spreadsheet version 12.3.1 and exported to R software version 3.5 for analysis.

Keds were collected longitudinally for a 3-year period spanning wet, late wet, and dry seasons. To determine whether the observed ked infestation on camels (response variable) and the likelihood of pathogen transmissions was influenced by environmental (seasonal conditions; dry, wet, late wet) and host-specific factors (age; young or mature) or sex (male, female), we fitted a generalized linear model (*glm*) assuming a Poisson error distribution with *log* link function. The influence of the following covariates was further analysed; exposure frequency and ked average numbers on pathogen transmission status (positive = 1, negative = 0) of experimental mice. Differences between various variables were compared using analysis of deviance F statistics with Chi-square test. For all the analyses, a *p* value of less than 5% ($p < 0.05$) was considered significant. ANOVA was used to determine variations in ked infestation between livestock species. The choice of these statistical analyses was informed by the random infestation of keds on hosts recorded as counts, and the assumption that their variances were equal to means. Additionally, we modelled the data using *glm*() *function* to determine how varying ked numbers on hosts influenced disease transmission among the sampled animals.

All study nucleotide sequences were edited in Geneious Prime software version 2019.1.1 (created by Biomatters) using the MAFFT plugin [36] and aligned with related sequences identified by querying the GenBank nr database using the Basic Local Alignment Search Tool (www.ncbi.nlm.nih.gov/BLAST/).

Phylogenetic and molecular evolutionary analyses were performed using Geneious Prime software version 2019.1.1. Maximum-likelihood phylogenies were constructed using PhyML v. 3.0 with automatic model selection based on the Akaike information criterion with 1000 bootstrap replicates [37]. A *Wolbachia* endosymbiont (family *Anaplasmataceae*; GenBank accession: KJ814215) was included in the phylogenetic tree as outgroup. Phylogenetic trees were visualised using FigTree v. 1.4.4 (http://tree.bio.ed.ac.uk/software/figtree/)

## Results

### Survey of camel keds, *Hippobosca camelina*, in northern Kenya

(a) **Ked infestations in different herds.** Identification of keds collected from camels, sheep, goats, cattle, dogs, and donkeys showed that *H. camelina* was predominantly found on camels, suggesting that *H. camelina* (Fig 2) is camel-specific. We collected up to a maximum of three *H. camelina* flies in a few mixed goats and sheep herds that were co-herded with camels. Despite co-herding of livestock species with camels, *H. camelina* was not found on cattle, dogs, and donkeys. Preliminary findings from our ongoing studies aiming to identify livestock keds suggest occurrence of different species of keds on sheep, goats, cattle, dogs, and donkeys, except in camels that were infested by only *H. camelina*.

The highest mean total ked numbers were found on camels, with fewer infesting flies on sheep and goats, significantly varying across the sampled animals (ANOVA, $F_{(5, 273)} = 12.64$, $p < 0.0001$). The mean counts of keds on camels were relatively high compared to those collected from cattle (Welch two-sample t-test; t = -7.2872, df = 206, $p < 0.001$), but not donkeys (t = 1.884, df = 16, $p = 0.078$). Similarly, more keds were collected from goats than sheep (t = 2.3442, df = 47, $p = 0.0234$). Moreover, we collected more keds on cattle in comparison to sheep (t = -5.4735, df = 58, $p < 0.001$). No significant variation was found between mean ked counts on donkeys and dogs (t = 0.8346, df = 15, $p = 0.4174$), and between camels and dogs (t = 0.4602, df = 11, $p = 0.6545$) (Fig 3A).

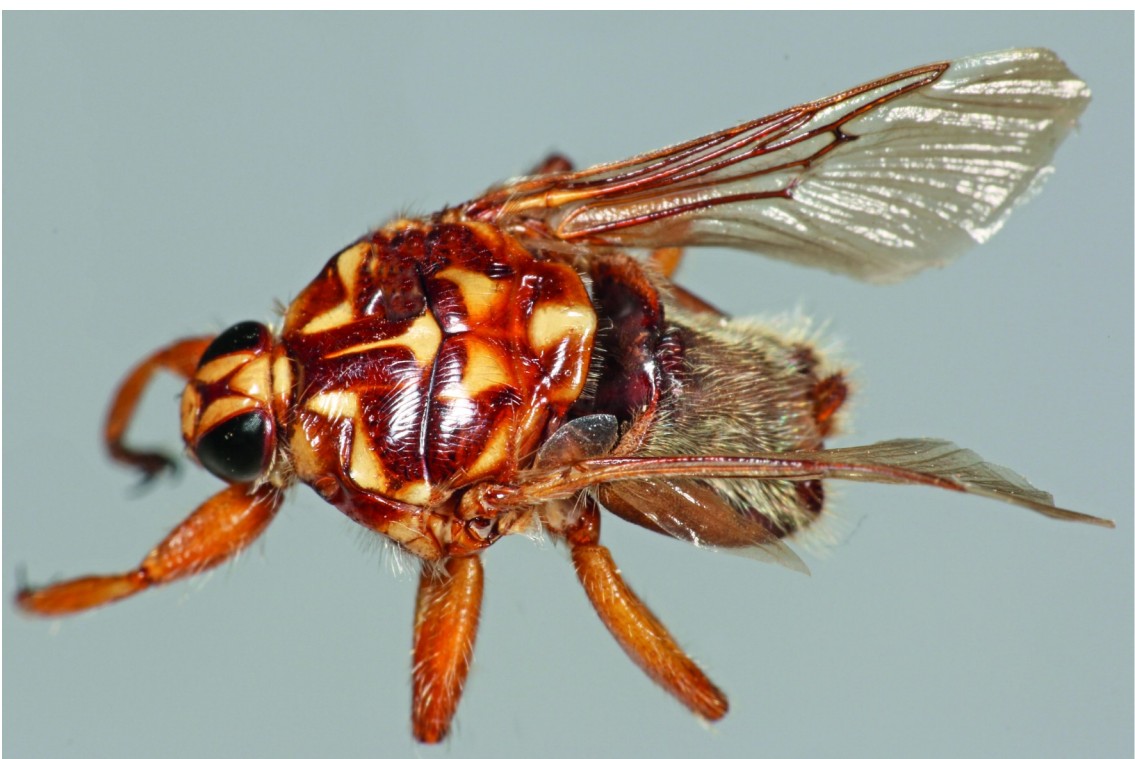

**Fig 2. *Hippobosca camelina*.** *Hippobosca camelina* was present: (i) in all of the camel herds surveyed, (ii) throughout the year, and (iii) nearly all camels in the herds were infested by keds.

**(b) Keds infestation on camels of different ages.** Ked infestation significantly varied with the camel age; young camels under two years of age were infested by 1–35 keds/camel with an average of nine flies per camel, whereas camels older than 2 years were infested by 0–82 keds/camel with a mean of 13 flies/host (glm; $F_{(1,528)} = 4375$, $p < 0.001$). In contrast, the number of camel keds did not differ significantly with camel sex (glm; $F_{(1,527)} = 4374$, $p = 0.5508$) (Fig 3B).

**(c) Seasonal variations of ked infestations on camels.** The average number of keds varied significantly with seasons (glm; $F_{(1,699)} = 7835$, $p < 0.001$). The highest number of keds were present on camels in the middle of dry season, reaching up to 100 keds/camel. In contrast, during the wet season, the number of keds/camel reduced to an average of seven keds/camel ($n = 14$ camel herds)., Camel keds numbers increased gradually to an average of 10 keds/camel towards the late wet season in June-July 2018. This rising trend in the number infesting camel keds continued until reaching peak numbers of 80–100 keds/camel in the dry season (Fig 3C).

**(d) Proportion of male and female keds in dry, wet, and late wet seasons.** We found more female than male keds irrespective of the season of the year in all randomly collected keds from camel herds in different geographical locations sampled at different times. The ked collections comprised of twice as many female flies than males in dry season ($n = 222$ keds) (Fig 3D). This 2:1 ratio of female:male keds was consistent in all seasons (ANOVA, $F_{(1,4)} = 0.1024$, $p = 0.765$), (wet season, $t = 4.4231$, df = 1, $p = 0.1416$; dry season, $t = 3$, $p = 0.2048$; late-wet season, $t = 7.0285$, df = 1, $p = 0.08997$).

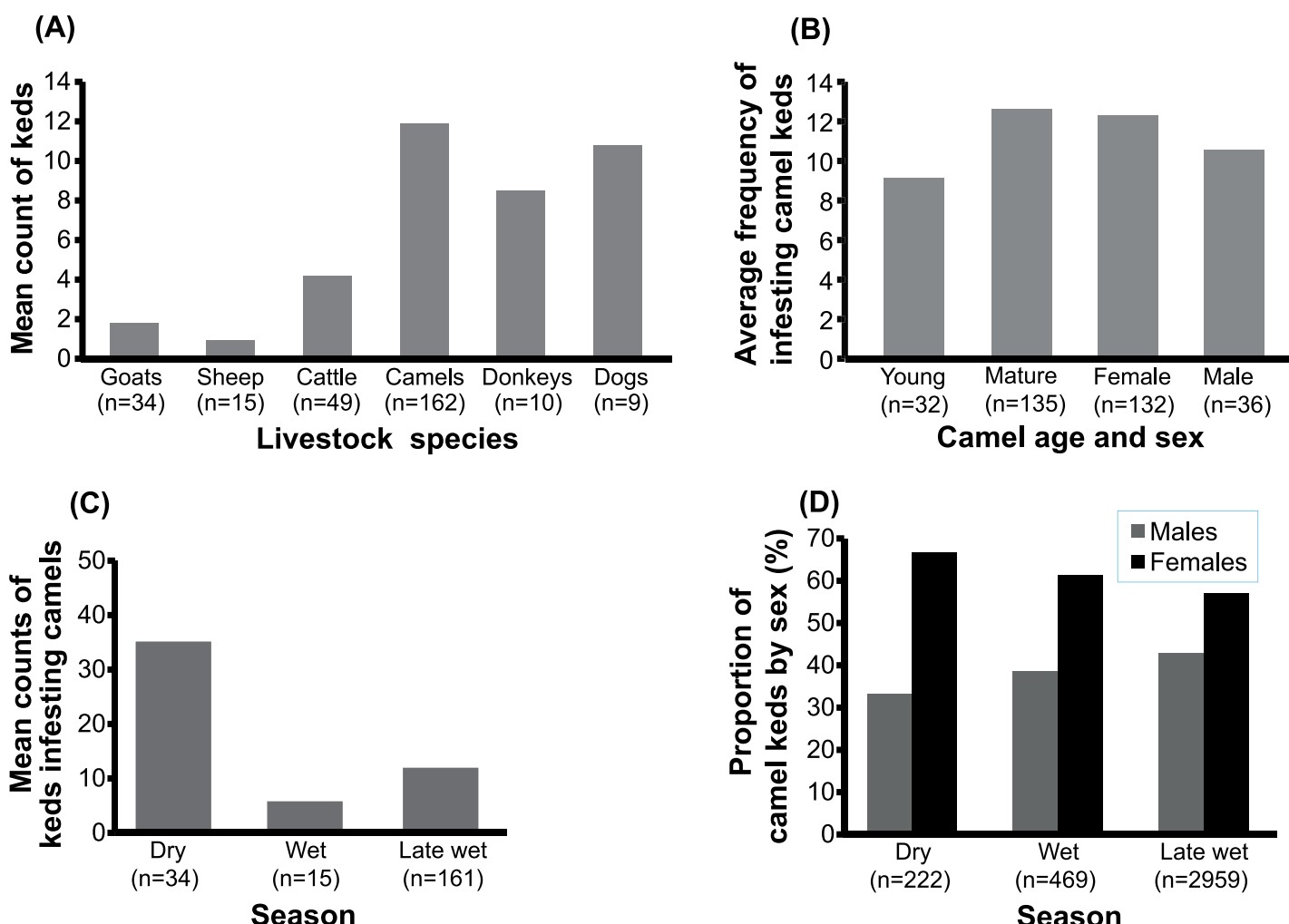

**Fig 3. Baseline survey data of livestock keds in Laisamis, northern Kenya. (A) Ked infestation in domestic animals.** *Hippobosca camelina*, that spends its adult life attached on the host, was predominantly found on camels, indicating its camel-specific preference. However, in rare instances, about 1 to 3 *H. camelina* flies were collected from sheep and goat herds that were co-herded with camels (but not found on other livestock species studied here). Highest mean ked infestation was recorded on camels, followed by dogs, and then donkeys. **(B) The influence of host age and sex on the preference of keds to infest camels.** Mature camels had higher average numbers of keds as compared to the young camels. Camel sex did not influence ked infestation ($p$ = 0.5508). **(C) Seasonal variations of ked infestations on camels.** The highest numbers of keds were recorded on camels during dry season followed by late wet season. The lowest ked numbers were recorded on the camels during the wet season. **(D)Proportions of male and female keds on camels.** The proportion of female keds was higher than the male flies sampled across the three seasons.

### Molecular detection of *Anaplasma* spp. in camels and their keds

**(a)** ***Anaplasma* in camels.** The prevalence of *Anaplasma* infection in camels was generally high throughout the year. The rate of infection during dry, wet, and late wet seasons was 70.3% ($n$ = 175/249), 63.9% ($n$ = 179/280), and 77.9% ($n$ = 348/447), respectively (Table D in S1 Text).

**(b)** ***Anaplasma* in camel keds.** The prevalence of '*Ca.* Anaplasma camelii' in keds varied with season, with the percentage prevalence being: 9.9% (dry), 20.8% (wet), and 28.9% (late wet) (Table D in S1 Text).

### Experimental transmission of *Anaplasma* from camels to mice and rabbits

The percentage red blood cells containing *Anaplasma* was 1.6% in camels (Fig 4A), 2.7% in mice (Fig 4B), and 3.0% in rabbits (Fig 4C), respectively (Fig 4). This *Anaplasma* sp. was

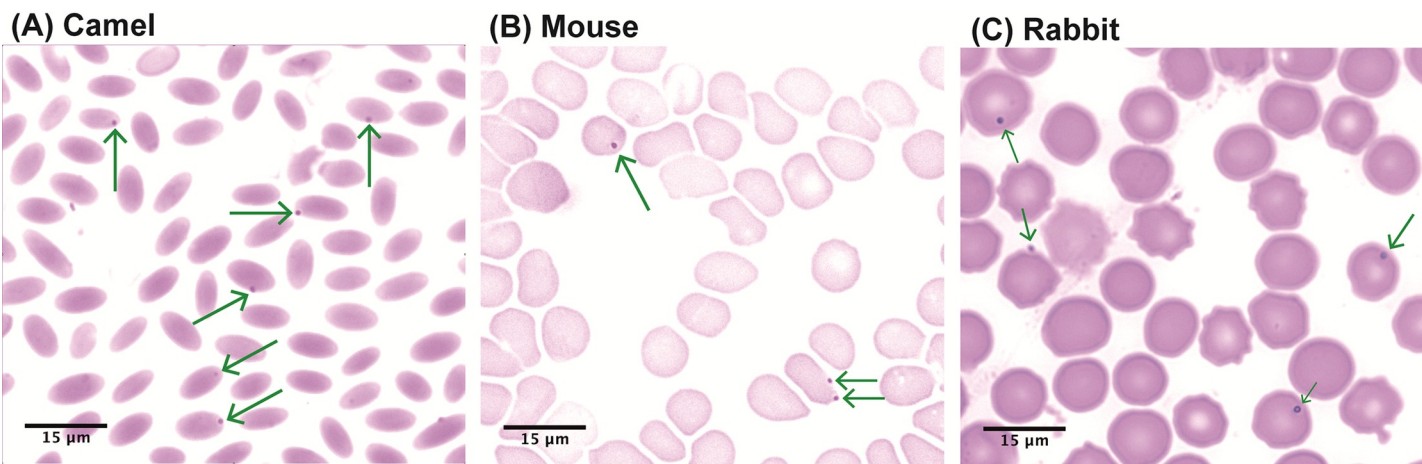

**Fig 4. Field's staining showing *Anaplasma* sp. infections in representative thin film blood smears. (A)** Naturally infected dromedary camel, **(B)** Experimental mouse exposed to ked bites, **(C)** Experimental rabbit post-ked bites. Green arrows point to *Anaplasma* sp., magnification x100.

localized at the periphery of infected erythrocytes and appeared in dot-like forms similar to the morphology of *Anaplasma marginale*.

**(i) Molecular analysis.** *Anaplasma* species was amplified in the test animals by PCR-HRM generating HRM melt curves (Fig 5).

The prevalence of *Anaplasma* infection in the test mice for the three repeat experiments was 47.4%, 6.9% and 17.9%, respectively, and 25% in rabbits (Table 2).

**(ii) Identification of *Anaplasma* spp. in test animals by gene sequencing.** PCR amplicons of representative positive test samples were sequenced using 1000-bp 16S rRNA and the shorter 300-bp 16S rRNA gene markers. Multiple sequence alignment of ~1000-bp 16S rRNA *Anaplasmataceae* sequences obtained from camels and mouse shared 100% identity to '*Ca*. Anaplasma camelii' (S1 Fig). Alignment of shorter *Anaplasmataceae*-specific 300-bp 16S rRNA sequences showed sequence identity of 100% between the sequences derived from experimental mice and rabbits that were exposed to repeated ked bites. The identity of *Anaplasma* sp. was confirmed to be '*Ca*. Anaplasma camelii' and there was no case of mixed *Anaplasma* species infection in all *Anaplasma*-positive samples (camels, camel keds, mice, and rabbits).

All sequences generated in this study were submitted to the GenBank with accession numbers as follows; (i) Camel keds: MK754149-MK754151 and MT510535-MT510537, (ii) Camels: MK754152-MK754154, MT510527, MT510529, MT510531, MT510532, and MT510534, (iii) Mice: MK754155-MK754160 and MT510538, and (iv) Rabbit: MT510539. The GenBank accessions for the longer ~1000-bp sequences used for phylogenetic analysis include; MK388294-MK388300, MT510528, MT510530 and MT510533 (sequenced in camel) & MK388301 (sequenced in test mouse after ked feeding bites).

Phylogenetic investigation of sequences obtained from this study with related sequences from GenBank showed that they are genetically identical to '*Ca*. Anaplasma camelii' previously sequenced from camels in Kenya (GenBank accession numbers: MN306315, MH936009 and MT929169), Saudi Arabia (KF843824) and Iran (KX765882) and closely related to *Anaplasma platys* [10,38] (Fig 6).

## Ked-feeding bite exposure frequencies influenced transmission of '*Ca*. Anaplasma camelii'

The success of '*Ca*. Anaplasma camelii' transmission to the experimental mice and rabbits by camel keds during blood feeding was influenced by the exposure frequencies to fly bites

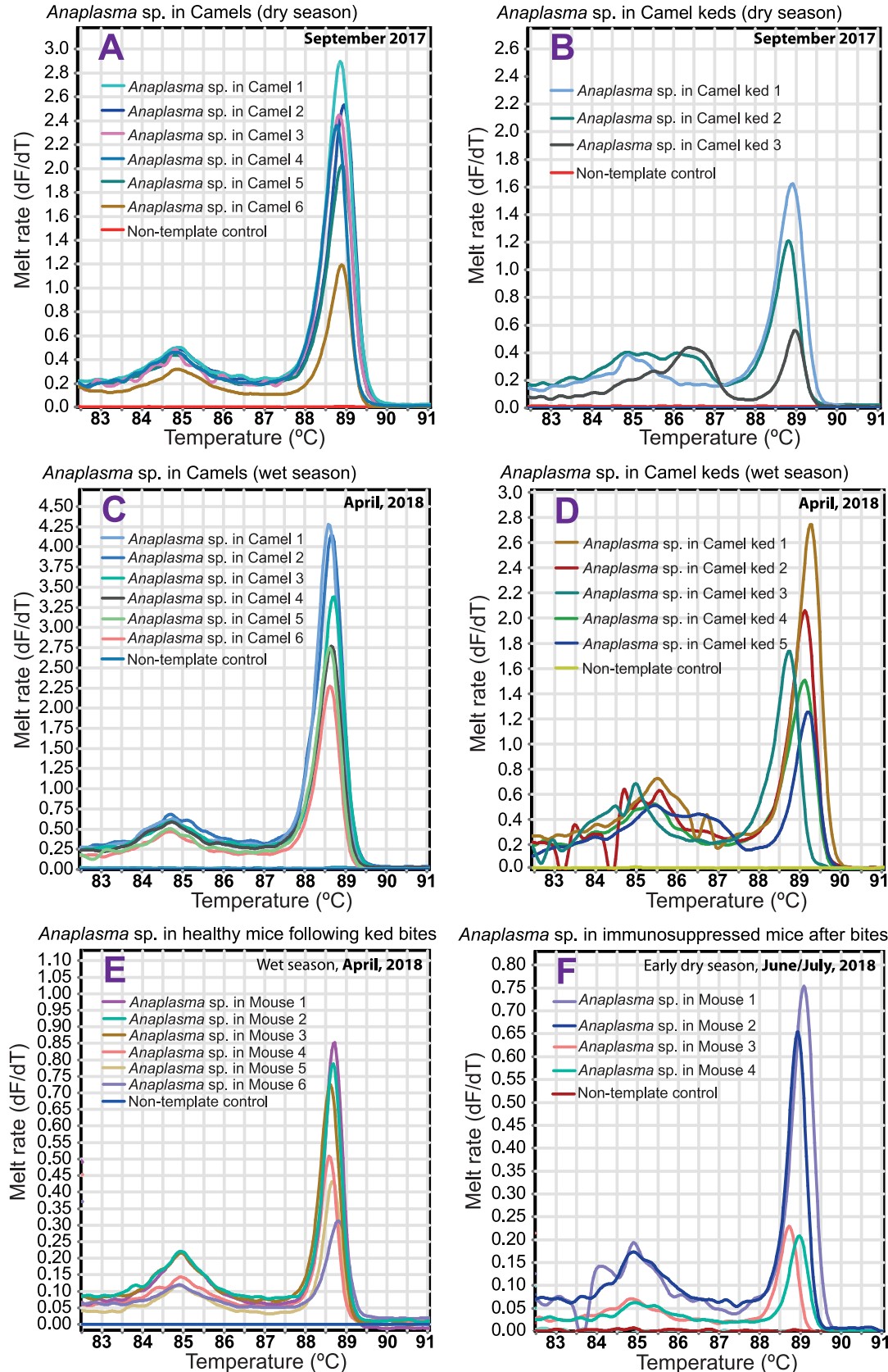

**Fig 5. PCR—HRM melt curves for detection of *Anaplasma* sp. in mice, camels, and keds.** DNA sequencing identified *Anaplasma* sp. in camels (A & C), keds (B & D), and mice (both healthy- E and immunosuppressed mice- F) as '*Ca*. Anaplasma camelii'.

(OR = 0.6504 (95% CI = 0.4714–0.8373), $p$ = 0.0004, $n$ = 115) (Table A-C in S1 Text). The pathogen transmission success could also be influenced by the proportion of keds that harbored *Anaplasma* (Table E in S1 Text). We observed that in some cases, a single exposure lasting for 12 h was sufficient to transmit *Anaplasma* to experimental mice (Table A and B in S1 Text), whilst in other instances, repeated exposure frequencies of up to 10 times did not transmit *Anaplasma* (Table C in S1 Text). Our three repeat experiments to determine the vector competence of camel keds to transmit '*Ca*. Anaplasma camelii' from infected camels to healthy immunocompetent mice (Table A and C in S1 Text), immunosuppressed mice (Table B in S1 Text), and rabbits (Table C in S1 Text) were successful.

## Discussion

This study reports the first experimental transmission of '*Ca*. Anaplasma camelii' from naturally infected camels to laboratory animals, demonstrating their vector competence in transmitting *Anaplasma*. Whilst ticks are required for cyclic transmission of *Anaplasma* [6], co-occurrent infestation with keds could imply direct involvement of these hematophagous flies in efficient transmission and maintenance of this hemopathogen within and between camel herds in northern Kenya. We recorded high '*Ca*. Anaplasma camelii' infection rates of 63–78% in camels, and in contrast only 10–28.9% of camel keds carried *Ca*. Anaplasma camelii' DNA during the various seasons. The presence of *Anaplasma* DNA in or on the camel keds was not surprising because when they bite infected camel to acquire bloodmeal, they also inadvertently ingest the accompanying blood-borne pathogens [21].

Field's staining of thin blood smears prepared from camels, mice and rabbits, and examined to screen for presence of *Anaplasma* species revealed marginal occurrence of this pathogen in the host RBCs (Fig 4) similar to the well-studied *Anaplasma marginale*. Further, this *Anaplasma* sp. was identified as '*Ca*. Anaplasma camelii' by 16S rRNA gene sequencing. However, we were not able to amplify the longer ~1000-bp 16S rRNA gene sequences from keds, unlike in mouse and camels, suggesting that '*Ca*. Anaplasma camelii' does not multiply in the fly. This finding implies mechanical transmission of this *Anaplasma* species by the camel keds. Lower rates of *Anaplasma* prevalence in keds than in their camel hosts could suggest that in keds, '*Ca*. Anaplasma camelii' does not transform or change in developmental forms, nor do they increase in numbers, thus over time they die off naturally and flies may only acquire fresh bacteria through subsequent infected blood meals. These observations further support our hypothesis of mechanical transmission of *Anaplasma* sp. by keds, which is known to occur in other biting flies via interrupted blood feeding [8]. Further studies are needed to; (i) establish the location of '*Ca*. Anaplasma camelii' in keds, and (ii) study the fate of this *Anaplasma* sp. in camels, keds, and in mice and rabbits. The probability of *Anaplasma* transmission to the

**Table 2. Summary of *Anaplasma* infection in test mice and rabbits.**

| Experiment | Season | Host | *Anaplasma* infection prevalence in test animals |
|---|---|---|---|
| 1st | Dry, September 2017 | Mice | 47.4% ($n$ = 9/19) |
| 2nd | Wet, April 2018 | Mice | 6.9% ($n$ = 4/58) |
| 3rd | Late wet, July-August 2019 | Mice | 17.9% ($n$ = 22/123) |
| | | Rabbits | 25% ($n$ = 1/4) |

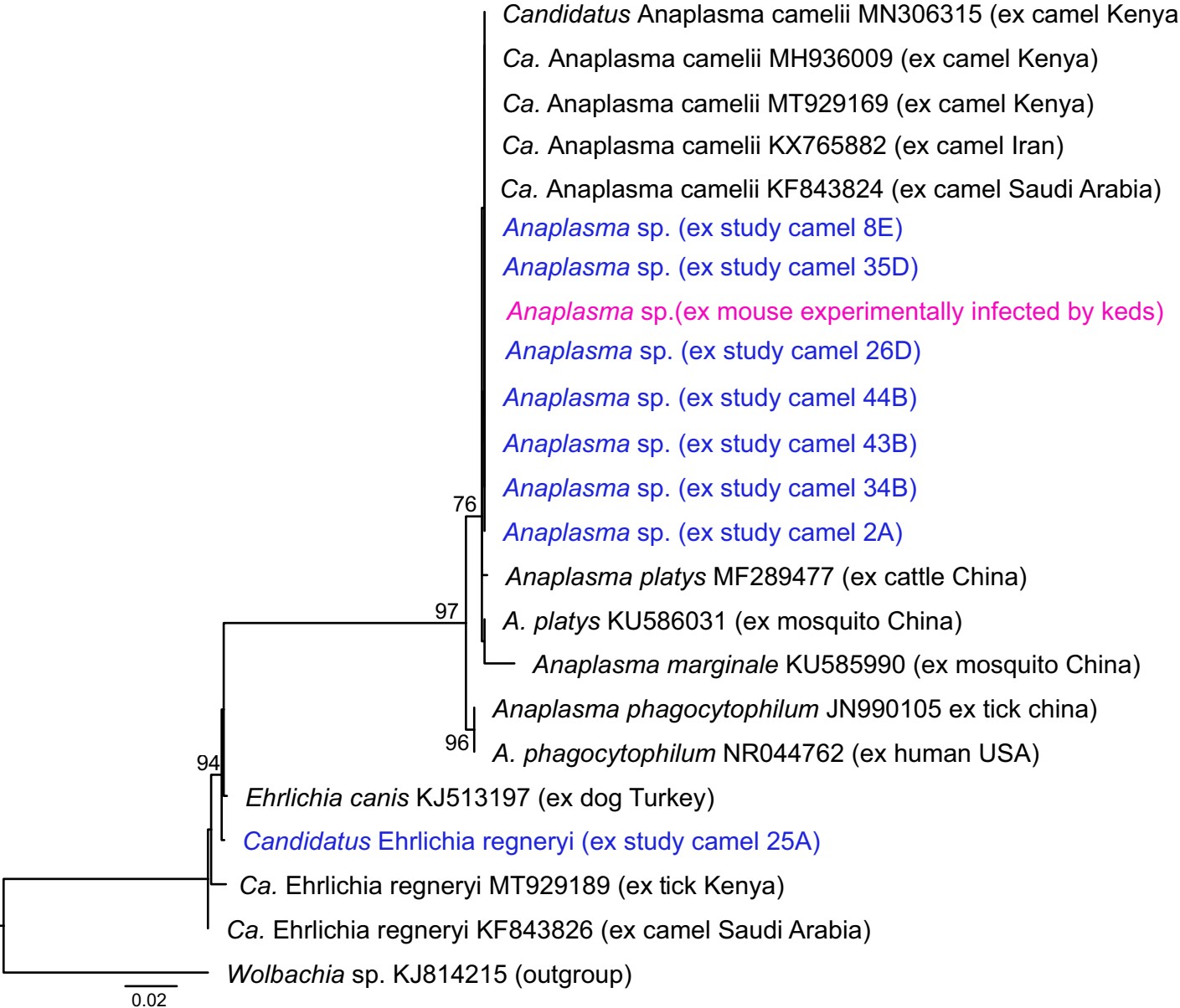

**Fig 6. Maximum-likelihood phylogenetic tree of 1000-bp *Anaplasma* sp. 16S rRNA sequences.** The sequences obtained from this study are shown in pink and blue colours and bootstrap values at the major nodes are the percentage agreement with 1000 bootstrap replicates. *Wolbachia* endosymbiont KJ814215 (family *Anaplasmataceae*) was included in the tree as outgroup.

experimental animals by keds increased with the frequency of fly bites. We also noted that the rates of contamination of keds with *Anaplasma* sp. varied in different geographical locations (Table E in S1 Text) and thus the proportion of contaminated keds that successfully bite the host to obtain blood meal could also influence the chances of pathogen transmission. We observed that the level of 'Ca. Anaplasma camelii' infection was higher in naturally infected camels than in experimentally infected mice and rabbits. Higher infection rates in camels could be attributed to infestation by other vectors such as ticks and *Stomoxys* that suck blood from camels and in the process spreading the *Anaplasma* pathogens. Also, keds could have better chances of mechanically transmitting *Anaplasma* pathogens among camels than in test

mice and rabbits because their feeding is more likely to be interrupted by the camels' response to painful fly bites, as opposed to restrained laboratory animals. Transmission of *Anaplasma* was achieved in both healthy and immunosuppressed mice, as well as in rabbits. However, smaller percentage of immunosuppressed mice acquired *Anaplasma* infection, possibly due to decreased fly numbers in the wet season of April 2018 that permitted only a maximum of three ked-mice exposure frequencies. We were unable to determine *Anaplasma*-positivity rates in freshly collected live camel flies used in pathogen transmission experiments using mice and rabbits.

*Anaplasma* 16S rRNA sequences obtained from camel, mouse, and rabbit showed 100% sequence identity as shown by multiple sequence alignment (S1 Fig). BLAST searches of *Anaplasma* sequences from mouse and rabbit revealed 100% identity to '*Ca*. Anaplasma camelii' sequenced from camels, suggesting naturally-infected camel hosts as the common origin of this pathogen. The infection mechanism, its main host target cells, and pathogenicity of '*Ca*. Anaplasma camelii' in camels is not understood at present. Research studies using *in vitro* pathogen cultures in tick cell lines [39] will provide knowledge on infection mechanism of this *Anaplasma* species.

The occurrence and abundance of keds on vertebrate hosts throughout the year could enhance transmission and maintenance of *Anaplasma* pathogens in camels. We found the number of female camel keds to be almost twice that of males in random fly collections. Earlier studies on *Hippobosca equina* reported similar findings on sex ratios that were biased towards more females than males [40]. This is presumably important in maintaining ked populations to compensate for their low reproduction rates.

Further, our studies show that *H. camelina* is camel-specific because it largely infested camels, but was hardly found on the other co-herded livestock species such as sheep, goats, cattle, donkeys, and dogs. Camels suffer the most burden from biting keds with greater infestations than all other co-herded livestock species, therefore making them most vulnerable to ked-borne diseases. Host selection by keds is possibly achieved through visual and olfactory signals as reported in other dipterans such as tsetse fly [41–44].

The highest ked count of up to 100 keds/camel occurred in the dry season in September 2017, whereas the lowest mean count of seven keds/camel was recorded in the wet season in June-July 2018. Then, towards the late wet season, the fly numbers on camels gradually increased reaching highest fly numbers in September of the following dry season. Thus, the number of camel-infesting keds sharply decreased in wet season, in contrast to the dry season. Starved camel adult keds that departed from their vertebrate hosts will actively search for nearby vertebrate hosts, that include domestic and wild animals, as well as humans [45] and possibly spreading diseases to them.

Temperatures of about 32°C and relative humidity of 75% are key to successful pupal development in *H. equina* and *H. camelina* [46]. However, since wet seasons occur once or twice in a year in northern Kenya, it is reasonable that these keds, which have a low reproduction rate like their tsetse relatives, must continue to larviposit throughout the year to sustain their populations. Gravid females of *H. equina* deposit a single 3rd instar larva every 3–8 days in the cracks of mud walls of stables [46]. During our field studies, we were unable to locate the larviposition sites of gravid *H. camelina*. Better understanding of the life cycle and biology of camel keds could enhance the design of fly control strategies. There are no ked-specific traps that are currently available for controlling these potential vectors of zoonotic diseases. Tsetse fly traps, such as NGU traps designed at *icipe*, work on the basis of visual (blue-black color of the trap clothing) and olfactory cues (cow urine and acetone as attractants) can also trap biting flies such as *Stomoxys* and *Tabanus* species, but rarely camel keds as monoconical traps could only catch 0–3 keds per trap/day [21]. It is likely that keds perceive colors and olfactory cues

differently, unlike tsetse fly, their closest relatives; hippoboscids and tsetse flies belong to the same superfamily, *Hippoboscoidae*. It is important to understand how camel-specific *H. camelina* perceives their camel hosts via visual and/or olfactory cues in order to model ked-specific traps for targeting the unattached keds that are often found resting on vegetation. Control of *H. camelina* would be crucial in disease management and control for improved human and livestock health.

All camel keepers whose camels were sampled in this study also kept domestic animals including dogs and small ruminants. Co-herding of livestock promotes spread of vector-borne diseases. For instance, *A. platys* infection in dogs could be transmitted to camels [19] or to humans [13–15]. *Anaplasma phagocytophilum*, which commonly affects cattle, is also known to be pathogenic to humans causing human granulocytic anaplasmosis [47–49]. 'Candidatus Anaplasma camelii' was previously reported in dromedary camels in Kenya [21,50,51], Saudi Arabia [10], Iran [38], and Morocco [12]. Tick vectors belonging to genus *Hyalomma* were suspected to transmit '*Ca*. Anaplasma camelii' in Moroccan one-humped camels that are reportedly the reservoir host of *Anaplasma* spp. [12].

Our study was limited by short sampling periods lasting between 7–30 days that limited pathogen transmission studies, which relied on the field-collected keds, as we could not establish long-term laboratory colonies of *H. camelina* due to high mortality of flies under the controlled laboratory conditions. This limited the frequencies of ked bites on the test animals, possibly reducing pathogen infection rates. Additionally, seasonal variation of ked populations also affected the transmission experiments. For instance, our experiment in April 2018, during wet season, was limited by very low ked numbers on camels, thus we could not keep ked exposure rates constant in all repeat experiments.

In conclusion, we demonstrate for the first time the ability of camel-specific keds, *H. camelina*, to transmit '*Ca*. Anaplasma camelii' originating from naturally infected dromedary camels to laboratory-reared mice and rabbits through blood-feeding bites. The prevalence of camel anaplasmosis caused by a single *Anaplasma* sp., '*Ca*. Anaplasma camelii', was high throughout the seasons, unlike in keds, which had low rates of contamination. Despite these low rates of '*Ca*. Anaplasma camelii' contamination, keds were still able to transmit pathogens in all test groups of mice and rabbits. Further studies are needed to determine the vector competence of keds in transmission of other blood-borne pathogens of veterinary and zoonotic importance.

## Supporting information

**S1 Text. Table A in S1 Text: The feeding schedule of camel keds, *Hippobosca camelina*, on healthy Swiss white mice for pathogen transmission experiment.** Detection of '*Ca*. Anaplasma camelii' in experimental mice group was determined post-ked bites by PCR-HRM using genus-specific primers for 16S rRNA gene target. The data shows that 47.4% of mice in the test group (*n* = 9/19) have acquired *Anaplasma* infection following ked blood-feeding bites. The control mice group (*n* = 2) was not exposed to the biting flies. **Table B in S1 Text: The feeding schedule of camel keds, *H. camelina*, on immunosuppressed mice (*n* = 60) for *Anaplasma* transmission to determine the effect of immunosuppression on mice infection.** PCR-HRM analysis revealed *Anaplasma* infection rate of 6.9% in test mice (*n* = 4/58) two weeks post-ked bite exposure. Control mice were not exposed to fly bites. We recorded *Anaplasma* infection rate of 12.9% (*n* = 4/31) in mice after 60 days of follow-up screening, but all 27 mice that were sacrificed 140 days post-ked exposure were not infected with the *Anaplasma* sp. **Table C in S1 Text: The feeding schedule of camel keds, *H. camelina*, on mice (*n* = 123) and rabbits (*n* = 6) for *Anaplasma* transmission study to determine vector competence of keds.** PCR-HRM analysis targeting genus-specific 16S rRNA gene detected *Anaplasma*

infection rates of 17.9% in test mice ($n$ = 22/123) and in 25% of rabbits ($n$ = 1/4) post-ked bites. Control mice ($n$ = 8) and rabbits ($n$ = 2) were not exposed to biting flies. **Table D in S1 Text: The infection prevalence of '*Ca*. Anaplasma camelii' in camels and camel keds in various seasons of the year; wet, late wet, and dry season. Table E in S1 Text: Presence of *Anaplasma* sp. in camel keds collected from camel herds in various geographical locations in Laisamis, northern Kenya.**
(DOCX)

**S1 Fig. Multiple sequence alignment of eight ~1000-bp *Anaplasma* 16S rRNA genes.** One sequence from the test mouse (3A_M_C) and seven sequences, 8E - 35D, from naturally infected camels were 100% identical (MUSCLE v3.8) and BLAST of GenBank showed highest identity of 100% to '*Ca*. Anaplasma camelii'.
(DOCX)

## Acknowledgments

We are grateful to camel farmers in Laisamis for allowing us to collect samples. Many thanks to Galtumo Lordagos, Ogoga Kaldale, Lmachungwan Galgitele, Ltulusuan Letoiye, Moika Naparakwo, Fereiti Bargul, and Ali Baltor for assisting in restraining camels during collection of blood and ked samples. John Ng'iela of *icipe*'s Animal Health Theme is acknowledged for providing valuable support during field sampling, staining and interpreting Field's stained blood smears for identification of *Anaplasma* spp. We are thankful to Nigel Wyatt of the Natural History Museum in London and Edgar Turner of the Zoology Museum of the University of Cambridge for their help during morphological identification of keds. Winny Cherono, currently a graduate member at the Institute of Surveyors of Kenya (GIS Chapter), generated maps for this study. We are also very grateful to Collins Kigen for his technical support.

## Author Contributions

**Conceptualization:** Joel L. Bargul, Mark Carrington, Daniel K. Masiga.

**Formal analysis:** Joel L. Bargul, Kevin O. Kidambasi, Merid N. Getahun, Jandouwe Villinger, Robert S. Copeland, Jackson M. Muema, Mark Carrington, Daniel K. Masiga.

**Funding acquisition:** Joel L. Bargul, Mark Carrington, Daniel K. Masiga.

**Investigation:** Joel L. Bargul, Kevin O. Kidambasi, Robert S. Copeland.

**Methodology:** Joel L. Bargul, Kevin O. Kidambasi, Robert S. Copeland, Mark Carrington, Daniel K. Masiga.

**Project administration:** Joel L. Bargul.

**Resources:** Joel L. Bargul, Merid N. Getahun, Jandouwe Villinger, Robert S. Copeland, Mark Carrington, Daniel K. Masiga.

**Supervision:** Joel L. Bargul, Merid N. Getahun, Jandouwe Villinger, Robert S. Copeland, Mark Carrington, Daniel K. Masiga.

**Validation:** Joel L. Bargul, Kevin O. Kidambasi, Robert S. Copeland, Mark Carrington, Daniel K. Masiga.

**Visualization:** Joel L. Bargul, Kevin O. Kidambasi.

**Writing – original draft:** Joel L. Bargul.

**Writing – review & editing:** Joel L. Bargul, Kevin O. Kidambasi, Merid N. Getahun, Jandouwe Villinger, Robert S. Copeland, Jackson M. Muema, Mark Carrington, Daniel K. Masiga.

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
