## [Decision Letter · Decision Letter 0]

16 May 2021

Dear Dr. Bargul,

Thank you very much for submitting your manuscript "Transmission of ‘Candidatus Anaplasma camelii’ to laboratory animals by camel-specific keds, Hippobosca camelina" for consideration at PLOS Neglected Tropical Diseases. As with all papers reviewed by the journal, your manuscript was reviewed by members of the editorial board and by several independent reviewers. The reviewers appreciated the attention to an important topic. Based on the reviews, we are likely to accept this manuscript for publication, providing that you modify the manuscript according to the review recommendations. 

Dear Dr. Bargul and colleagues:

Thanks for submitting your manuscript to PLoS Neglected Tropical Diseases. I have now received two independent reviews of your work, and as you will see, the reviewers raised some minor concerns about the research (mostly the manuscript format and content, although clarity is needed on experimental design in several cases). Despite this, these reviewers are optimistic about your work and the potential impact it will have on research studying vector biology and transmission dynamics of rickettsial species. Thus, I encourage you to revise your manuscript, accordingly, taking into account all of the concerns raised by both reviewers.

Please consider revising the title of your work to clearly indicate the scope/intent of the study. Also, please address the concerns regarding ked number and overall treatment of camels and why they specifically were investigated.

While the concerns of the reviewers are relatively minor, please ensure they are adequately addressed as the original reviewers will evaluate your responses to their concerns. There are not too many suggestions; thus, it should not take much effort to address these concerns to greatly improve your manuscript.

I look forward to seeing your revision, and thanks again for submitting your work to PLoS Neglected Tropical Diseases.

Good luck with your revision,

-joe

Sincerely,

Joseph James Gillespie, Ph.D.

Associate Editor

Job Lopez

Deputy Editor

Dear Dr. Bargul and colleagues:

Thanks for submitting your manuscript to PLoS Neglected Tropical Diseases. I have now received two independent reviews of your work, and as you will see, the reviewers raised some minor concerns about the research (mostly the manuscript format and content, although clarity is needed on experimental design in several cases). Despite this, these reviewers are optimistic about your work and the potential impact it will have on research studying vector biology and transmission dynamics of rickettsial species. Thus, I encourage you to revise your manuscript, accordingly, taking into account all of the concerns raised by both reviewers.

Please consider revising the title of your work to clearly indicate the scope/intent of the study. Also, please address the concerns regarding ked number and overall treatment of camels and why they specifically were investigated.

While the concerns of the reviewers are relatively minor, please ensure they are adequately addressed as the original reviewers will evaluate your responses to their concerns. There are not too many suggestions; thus, it should not take much effort to address these concerns to greatly improve your manuscript.

I look forward to seeing your revision, and thanks again for submitting your work to PLoS Neglected Tropical Diseases.

Good luck with your revision,

-joe

Reviewer's Responses to Questions

**Key Review Criteria Required for Acceptance?**

**Methods**

-Are the objectives of the study clearly articulated with a clear testable hypothesis stated?

-Is the study design appropriate to address the stated objectives?

-Is the population clearly described and appropriate for the hypothesis being tested?

-Is the sample size sufficient to ensure adequate power to address the hypothesis being tested?

-Were correct statistical analysis used to support conclusions?

-Are there concerns about ethical or regulatory requirements being met?

Reviewer #1: -Are the objectives of the study clearly articulated with a clear testable hypothesis stated? YES

-Is the study design appropriate to address the stated objectives? YES

-Is the population clearly described and appropriate for the hypothesis being tested? YES

-Is the sample size sufficient to ensure adequate power to address the hypothesis being tested? N/A

-Were correct statistical analysis used to support conclusions? YES

-Are there concerns about ethical or regulatory requirements being met? YES

Reviewer #2: 1.The authors have emphasized that camel flies are efficient fliers which is not the case. This statement is also contradicted by the same authors who have stated as follows in the same paper

• “Both male and female keds are obligate blood feeders that stay with their camel hosts, unless disturbed” ( lines 86-88).

• If camel flies hardly leave hosts unless disturbed, then what is the likelihood of them being transmitters of pathogens in nature? (lines 183-185) –

• Mechanical transmission requires that the vector frequently jumps from host to host; if these flies infrequently depart from their camel host then it is unlikely that they play such a role in nature. ( line 613-614)

2. Review questions to be addressed by the authors

• If H. camelina was found only on camels then what raw data was used to include other animals hosts in analysis as shown on Figure 3a. ?( Line 377-385).

• Is the comparision on proportion of keds depicted on Figure 3a based on H. camelina or different ked species?

• If H. camelina was found to be species specific on the camel host, then why was it important to consider collections from other animal hosts such as donkeys, cattle, goats , dogs and sheep?

3.Clarifications on reported information by the authors:

-(a)Authors state that rates of contamination of keds with Anaplasma sp. varied in different geographical location but this important statement is not supported with relevant data.(line 552-553).

- (b)A statement suggests that the number of keds did not reduce during the wet season as the keds simply shifted from camel hosts to rest on the vegetation near by.(line 595 to 598). This seems to contradict the findings that fly population numbers flactuated with seasons.

-(c)-The method used to count the camel keds in a seperate study to determine the seasonal fly populations is not specified.( line 191-193).

**Results**

-Does the analysis presented match the analysis plan?

-Are the results clearly and completely presented?

-Are the figures (Tables, Images) of sufficient quality for clarity?

Reviewer #1: -Does the analysis presented match the analysis plan? YES

-Are the results clearly and completely presented? YES

-Are the figures (Tables, Images) of sufficient quality for clarity? YES

Reviewer #2: Accept:

Data analysed matches the experimental plan and design

Results are well presented with detailed Tables and figures

All Tables and Figures are of high quality

**Conclusions**

-Are the conclusions supported by the data presented?

-Are the limitations of analysis clearly described?

-Do the authors discuss how these data can be helpful to advance our understanding of the topic under study?

-Is public health relevance addressed?

Reviewer #1: -Are the conclusions supported by the data presented? YES but could be refined after attending to suggested changes

-Are the limitations of analysis clearly described? yes

-Do the authors discuss how these data can be helpful to advance our understanding of the topic under study? partially: can be improved after attending to suggested changes 

-Is public health relevance addressed? Not applicable

Reviewer #2: Accept:

Conclusion are supported by the data presented

Limitations are well described

Data has been discussed well and will help understanding of the possible role of camel flies in the mechanical transmission of blood dwelling pathogens.

The public health relevance has been addressed in terms of food security

**Editorial and Data Presentation Modifications?**

Reviewer #1: Transmission of ‘Candidatus Anaplasma camelii’ to laboratory animals by camel-specific keds, Hippobosca camelina by Bargul et al. 

Reviewer’s summary

The authors set out to determine the seasonal variation in the density of camel-specific keds, Hippobosca camelina and other livestock keds infesting livestock in the Laisamis area of Marsabit county, northern Kenya. In addition, they purposed to experimentally determine if Hippobosca camelina can transmit Anaplasma species to laboratory animals [Rabbits and mice]; datasets they discuss herein as important in extending the knowledge of the readership of PNTD towards the understanding of the role of Hippobosca camelina in mechanical transmission of Anaplasma species to Camelus dromedarius populations and other domestic animals. This manuscript presents novel datasets and I deem it helpful in the management of camel Anaplasmataceae infections. This is a good manuscript in its area which would further be improved by attending to the following minor changes;-

1. This manuscript presents data on the seasonal variation in the density of camel-specific keds, Hippobosca camelina and other livestock keds infesting livestock in the Laisamis area of Marsabit county, northern Kenya but this is not reflected in the current title or key words. I would suggest that the authors consider improving their manuscript title and key words to reflect this; e.g “Seasonal variation in the density of camel-specific keds, Hippobosca camelina and demonstration of their experimental ability to transmission Candidatus Anaplasma camelii”.

2. In this MS, where the term “small laboratory animals” is used should be replaced with “mice and rabbits

3. Rephrase and /or improve language use for sentences in lines; 14-16, 45-49, 155-158, 189-193, 270-276, 393-395,534-536, 638-640, 644-647, 647-650

4. Line 202-203 and 233 …..with minor modifications; please state these minor modifications to allow for replication on the part of the reader

5. Line 219;…..Anaplasma transmission; please specify the species of Anaplasma 

6. Line 245; regularly….how regular? 

7. Lines 248… at intervals; please specify these intervals here

8. Line 286; details are provided for DNA extraction from Ked samples and not from camel , rabbit or mice blood samples. Please expand this section by providing details of how genomic/total DNA was extracted from the rest of the samples that were not keds

9. Data analysis: Lines 348-349. Rephrase this sentence and provide details [e.g packages/ software extension, interpolation distance etc] of how the maps were generated. 

10. Lines 350-351: Provide details of packages used and the statistics that were computed; probably explaining your choice

11. Lines 422-443, 452-455, 460-464, 474-481: The details in these lines are part of methods and not results. Please consider deleting these details or if needed transferring them to the right sections of methodology

12. Line 534; please specify the host whose blood smears were examined

13. Line 553 or there about; please explain here how level of Candidatus Anaplasma camelii in experimentally infected laboratory animals compares with that of naturally infected camels and how this comparison gives merit to the possibility of natural transmission of Candidatus Anaplasma camelii by Hippobosca camelina. 

14. Lines 554-559: to make the text presented in these lines valid, it would be good to explain why in experimentally testing immunity-based [immune status; suppression vs normal] factors as predictors of infection/transmission of Candidatus Anaplasma camelii, the authors did not keep exposure rate [number of keds] constant between groups! 

15. at Lines 590, 601 or there about, please discuss the likelihood of camel keds infesting or not infesting other livestock or even humans in the absence of camels; their preferred hosts. 

16. Lines 611-624: Please clarify why you would justify trap/target-based control methods for camel keds that spend > 10 hours attached and feeding on camels and not camel-based control methods e.g. insecticide application [Live bait technologies].

Reviewer #2: Accept

**Summary and General Comments**

Reviewer #1: All requested changes are detailed above

Reviewer #2: The paper is generally good with new findings;for example the presence of Anaplasma DNA in or on the camel keds is a novel finding. The work was done with adequate follow-up involving the local communities, who must have also learnt some thing new about the camel flies. The data is quite robust accompanied with detailed analysis. The presentation of Tables and Figures is of very high standards.

PLOS authors have the option to publish the peer review history of their article (what does this mean?). If published, this will include your full peer review and any attached files.

Reviewer #1: Yes: Dr Muhanguzi Dennis [PhD]

Reviewer #2: Yes: Prof Florence Awino Oyieke

Figure Files:

Data Requirements:

Reproducibility:

References

---

## [Editor Report · Decision Letter 1]

20 Jul 2021

Dear Dr. Bargul,

We are pleased to inform you that your manuscript 'Transmission of ‘Candidatus Anaplasma camelii’ to mice and rabbits by camel-specific keds, Hippobosca camelina' has been provisionally accepted for publication in PLOS Neglected Tropical Diseases.

Best regards,

Joseph James Gillespie, Ph.D.

Associate Editor

Job Lopez

Deputy Editor

Dear Dr. Bargul and colleagues:

Thanks for revising your manuscript based on the concerns raised by the reviewers. I now believe that your manuscript is suitable for publication. Congratulations! I look forward to seeing this work in print, and I anticipate it being an important resource for groups studying Anaplasma biology and vector biology. Thanks again for choosing PeerJ to publish such important work.

Best,

-joe

---

## [Editor Report · Acceptance letter]

11 Aug 2021

Dear Dr. Bargul,

We are delighted to inform you that your manuscript, "Transmission of ‘Candidatus Anaplasma camelii’ to mice and rabbits by camel-specific keds, Hippobosca camelina," has been formally accepted for publication in PLOS Neglected Tropical Diseases.

Best regards,

Shaden Kamhawi

co-Editor-in-Chief

Paul Brindley

co-Editor-in-Chief
